# Weight cycling and the subsequent onset of type 2 diabetes mellitus: 10-year cohort studies in urban and rural Japan

Hiroshi Yokomichi,[1] Sachiko Ohde,[2] Osamu Takahashi,[2] Mie Mochizuki,[3] Atsunori Takahashi,[1] Yoshioki Yoda,[4] Masahiro Tsuji,[4] Yuka Akiyama,[1] Zentaro Yamagata[1]

► Prepublication history and additional material are available. To view these files please visit the journal online (http://dx.doi.org/10.1136/bmjopen-2016-014684).

[1]Department of Health Sciences, University of Yamanashi, Chuo City, Yamanashi, Japan
[2]Center for Clinical Epidemiology, St. Luke's International University, Chuo Ward, Tokyo, Japan
[3]Department of Pediatrics, University of Yamanashi, Chuo City, Yamanashi, Japan
[4]Yamanashi Koseiren Health Care Center, Kofu City, Yamanashi, Japan

**Correspondence to**
Dr. Hiroshi Yokomichi;
hyokomichi@yamanashi.ac.jp

## ABSTRACT

**Objective** To investigate how weight cycling (gaining and losing weight) affects the risk of diabetes.

**Design** Cohort studies.

**Setting** Primary healthcare in urban and rural Japan.

**Participants** 20 708 urban and 9670 rural residents.

**Primary outcome measures** ORs for diabetes in those with weight loss, weight loss-gain, stable weight, weight gain-loss and weight gain over 10 years. Weight gain and loss were defined as a change of more than ±4% from baseline weight.

**Results** In the urban region, the ORs relative to the stable group for the loss-gain and gain-loss groups were 0.63 (95% CI 0.45 to 0.89) and 0.51 (95% CI 0.32 to 0.82) for men and 0.72 (95% CI 0.39 to 1.34) and 1.05 (95% CI 0.57 to 1.95) for women. In the rural region, they were 1.58 (95% CI 0.78 to 3.17) and 0.44 (95% CI 0.15 to 1.29) in men and 0.41 (95% CI 0.12 to 1.44) and 0.77 (95% CI 0.28 to 2.14) in women. The ORs for an increase in weight between 5 and 10 kg from the age of 20 years were 1.54 (95% CI 1.03 to 2.30) in men and 0.96 (95% CI 0.55 to 1.65) in women.

**Conclusions** In Japan, weight cycling was associated with a significant reduction in the risk of diabetes for men from urban regions. The associations were unclear for women from urban regions and both men and women from rural regions. These results differ from those in Western studies, probably because of differences in diet, insulin secretion and sensitivity and weight-consciousness.

## Strengths and limitations of this study

► Participants were invited from both urban and rural Japan.
► Several weight change patterns, including weight gain after loss and loss after gain, were measured.
► ORs may change with weight changes of >±4% in 10 years.
► The levels of insulin secretion and sensitivity were not measured.
► Whether participants' weight loss was intentional was undetermined.

## INTRODUCTION

Weight gain is a well-known risk factor for incidental type 2 diabetes. Research involving people of Western, Oriental and African descent has quantitatively established the risks of developing diabetes linked to weight gain.[1–3] Researchers have also raised the question of whether repeatedly gaining and losing weight (weight cycling) is an independent risk factor for developing diabetes. Studies on this topic have reported inconsistent results in Westerners in Europe and North America. Several prospective studies suggest that weight cycling is a risk factor for type 2 diabetes, but others do not.[4–8] To our knowledge, the risk of diabetes in Asian weight cyclers has not been researched.

There is a preconception linking being slim to an aesthetic standard,[9 10] and many Asian women therefore try to lose weight.[11 12] Another group likely to try to reduce their weight is middle-aged Asian businesspeople with few opportunities for physical activity. Recent studies have demonstrated that East Asians are much more likely than Westerners to develop type 2 diabetes at a lower body mass index (BMI).[13] It may therefore take only a small change in weight to alter the risk of diabetes for this group. The literature indicates that in Japan, diet, physical activity, prevalence of overweight individuals and aesthetic consciousness are different among urban and rural residents.[14 15] The aim of this study was to establish whether weight cyclers in Japan were at an increased risk of diabetes. We used Japanese urban and rural data to examine this question in populations with varying lifestyles.

## METHODS

### Study participants and measurements

These cohort studies involved participants from both an urban area, Tokyo and a rural area, Yamanashi Prefecture. In Tokyo, participants were employees of private companies

# Categorisation of weight change patterns

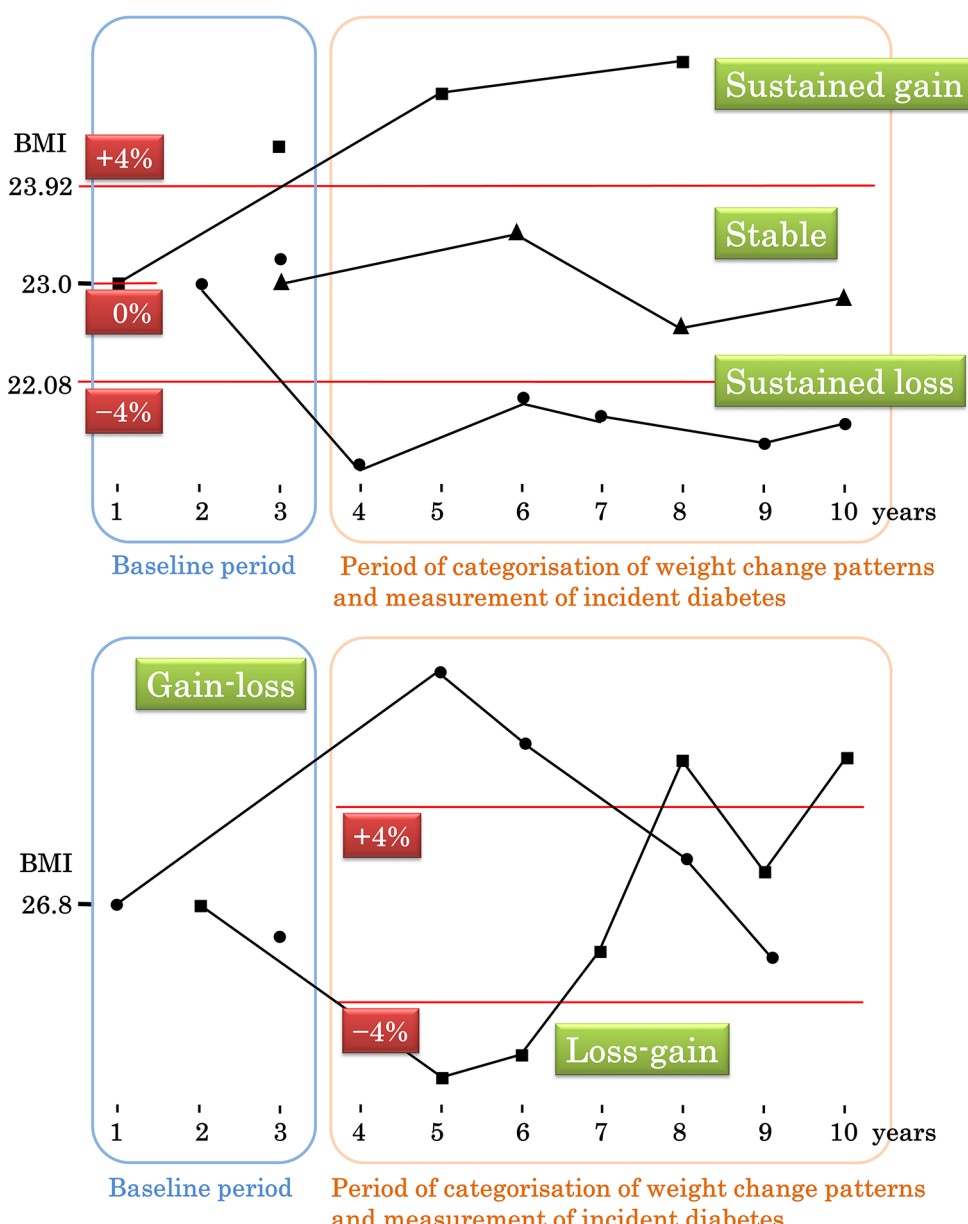

**Figure 1** How participants were categorised into the five weight change patterns. BMI, body mass index.

who underwent medical check-ups between January 2005 and December 2014 at St. Luke's International Hospital. These annual check-ups were based on a legal obligation imposed by the Industrial Safety and Health Act in Japan.[16] In Yamanashi Prefecture, participants were employees and also residents who paid for a private comprehensive medical check-up between April 1999 and March 2009 at the Yamanashi Koseiren Health Care Center. A subset of participants in Yamanashi used a subsidy from their employers or administrative agencies for this check-up. Those from the urban area therefore received approximately annual medical check-ups over a 10 year period, and those in the rural area received occasional voluntary check-ups. Participants were included

in the analysis if they had no diagnosis of diabetes and an HbA1c <6.5% (48 mmol/mol) during a baseline period of the first three years of the 10 year period. If they attended two or three medical check-ups in the first three years, the data from the first visit were used as the baseline. Participants were also required to attend at least two medical check-ups during the last seven years of the study. They therefore received 3–8 medical check-ups over the period to enable us to categorise weight change patterns (exposure). The onset of diabetes was indicated by the results of questions about the diagnosis of diabetes, the commencement of diabetic therapies or glycated haemoglobin (HbA1c)≥6.5% (48 mmol/mol).[17 18] In Tokyo, trained nurses interviewed participants over 20

years of age to establish changes in weight. BMI was calculated as the participant's weight in kilograms divided by the square of their height in metres.

### Weight change categories

The participants were categorised into five groups by their pattern of weight change during the 3–10 years after the baseline (figure 1). The stable group included all participants whose weight did not change by >±4% from the baseline. The sustained gain group consisted of those who gained >4% of their baseline weight and did not subsequently lose it again. The sustained loss group included those who lost >4% of their baseline weight and did not subsequently regain it. The gain-loss group included all participants who gained >4% of their baseline weight, but later brought their weight back <+4%. The loss-gain group included participants who lost >4% of their baseline weight but brought their weight back >−4%. When the participants had been categorised, we measured whether they developed diabetes. The length of time over which the participants were observed to see if they developed diabetes was between 1 and 6 years. Any data measured after a diagnosis of diabetes were ignored to conserve the temporality of exposure to outcome for epidemiological causation.[19]

The ±4% change in weight used for this categorisation was considered to be approximately a one-unit change for a person with a BMI of $22 \, \text{kg/m}^2$. This was based on the 2014 reference mean BMIs for Japanese men and women of 23.6 and $21.7 \, \text{kg/m}^2$.[20] The gain-loss and loss-gain groups were the weight cyclers of interest in this study.

### Statistical analysis

The baseline characteristics recorded for the participants included age, weight, height, BMI, HbA1c and fasting plasma glucose. We used univariate and multivariable logistic regressions to compare the risk of diabetes between the groups. In the data from the urban group, the covariates used for the adjustment at baseline were age, weight change from the age of 20, BMI, smoking habits, alcohol consumption and physical activity. In the data from the rural group, the available covariates for the adjustment at baseline were age, BMI, smoking habits and alcohol consumption. Means of follow-up duration and numbers of measurements for each weight change pattern group were calculated. The analyses were stratified by sex. Another focus of this study was the impact of weight cycling on incidental diabetes in middle-aged individuals. For a sensitivity analysis, we therefore restricted the analyses to a population aged 45–64 years. All statistical analyses used SAS statistical software (version 9.3, SAS Institute, NC, USA). The descriptive statistics were reported as the means and SD. All reported p-values were two-sided; p-values of <0.05 were considered statistically significant.

**Table 1** Baseline characteristics of the participants from urban (Tokyo) and rural (Yamanashi Prefecture) regions of Japan

| Characteristics, mean (SD) | Men | Women |
|---|---|---|
| Urban region | | |
| No. | 10 094 | 10 614 |
| Age, years | 49.6 (11.9) | 48.3 (11.3) |
| Weight, kg | 68.8 (9.7) | 52.3 (7.4) |
| Height, cm | 170.4 (6.1) | 157.7 (5.6) |
| Body mass index, kg/m$^2$ | 23.7 (2.8) | 21.0 (2.9) |
| HbA1c, % | 5.4 (0.3) | 5.3 (0.4) |
| HbA1c, mmol/mol | 35.3 (3.7) | 35.0 (3.8) |
| Fasting plasma glucose, mg/dL | 100.8 (8.6) | 94.5 (8.0) |
| Fasting plasma glucose, mmol/L | 5.6 (0.5) | 5.2 (0.4) |
| Rural region | | |
| No. | 4818 | 4852 |
| Age, years | 51.2 (10.3) | 52.1 (9.4) |
| Weight, kg | 65.7 (9.1) | 53.1 (7.5) |
| Height, cm | 168.1 (6.2) | 154.8 (5.6) |
| Body mass index, kg/m$^2$ | 23.2 (2.7) | 22.1 (2.9) |
| HbA1c, % | 5.3 (0.3) | 5.3 (0.3) |
| HbA1c, mmol/mol | 34.3 (3.8) | 34.4 (3.5) |
| Fasting plasma glucose, mg/dL | 96.3 (9.0) | 93.2 (8.5) |
| Fasting plasma glucose, mmol/L | 5.3 (0.5) | 5.2 (0.5) |

## RESULTS

### Participants

In total, 10 094 men and 10 614 women were enrolled from the Tokyo urban area, and 4818 men and 4852 women from Yamanashi Prefecture. Their baseline characteristics are shown in table 1 and are described online as supplementary table for details.

### Risk of diabetes in urban and rural Japan

Table 2 shows means (SDs) of follow-up duration and numbers of measurements in each weight change group. The numbers of measurements were smallest in the stable group in both sexes and both regions. Tables 3 and 4 show the incidence of diabetes and the odds ratios (ORs) for each explanatory variable in the urban and rural regions. Table 5 shows the risk of diabetes linked to weight cycling in the middle-aged population. In the urban data, the ORs were adjusted for baseline age and BMI, weight change from 20 years of age, smoking and drinking habits, amount of walking per day and sessions of physical activity per week; and in the rural data, the ORs were adjusted for baseline age, BMI and smoking and drinking habits.

## DISCUSSION

The data for the Japanese urban region suggest that the risk of diabetes in male weight cyclers was significantly lower than in those who maintained a stable weight (table 3). The diabetes risk for female weight cyclers in the urban region was non-significantly lower than or similar to the risk for those maintaining a stable weight. The data for the rural region suggest no difference in diabetes risk for weight-cycling men and a non-significantly lower risk for women compared with maintaining a stable weight (table 4). These results are reinforced by a sensitivity analysis focusing on middle-aged individuals, yielding almost identical non-significant ORs (table 5).

These observations are not consistent with those from previous studies of Western populations.[4-6 21] These studies showed that weight cycling significantly or non-significantly increased the risk of diabetes for point estimates. In the Framingham Heart Study, approximately $1 \text{ kg/m}^2$ of weight cycling in middle-aged Americans carried a hazard ratio (HR) of 1.1 (95% CI 0.8 to 1.5) for the risk of diabetes after adjusting for sex and BMI at 25 years of age.[4] In American middle-aged women examined in the National Health and Nutrition Examination Survey, weight cycling of 4.5–9.1 kg and 9.1–22.2 kg with

| Table 2 | Means (SD) of follow-up duration and numbers of measurements in the groups of weight change patterns | | |
|---|---|---|---|
| | Weight change pattern over 10 years | Follow-up duration, years | Measurements (n) |
| *Urban region* | | | |
| Men | Sustained loss | 7.7 (1.9) | 5.8 (1.6) |
| | Loss-gain | 8.2 (1.2) | 5.8 (1.4) |
| | Stable | 7.9 (1.7) | 4.9 (1.7) |
| | Gain-loss | 8.2 (1.2) | 5.7 (1.4) |
| | Sustained gain | 7.4 (1.9) | 5.4 (1.6) |
| Women | Sustained loss | 7.9 (1.7) | 6.0 (1.5) |
| | Loss-gain | 8.2 (1.2) | 5.9 (1.4) |
| | Stable | 7.9 (1.6) | 5.0 (1.7) |
| | Gain-loss | 8.2 (1.1) | 5.9 (1.4) |
| | Sustained gain | 7.5 (1.9) | 5.6 (1.6) |
| *Rural region* | | | |
| Men | Sustained loss | 7.3 (1.7) | 5.2 (1.6) |
| | Loss-gain | 7.9 (1.4) | 6.2 (1.2) |
| | Stable | 7.1 (1.9) | 5.2 (1.6) |
| | Gain-loss | 7.8 (1.4) | 6.1 (1.3) |
| | Sustained gain | 7.4 (1.6) | 5.4 (1.6) |
| Women | Sustained loss | 7.4 (1.7) | 5.4 (1.5) |
| | Loss-gain | 7.9 (1.3) | 6.1 (1.3) |
| | Stable | 7.0 (1.8) | 5.0 (1.6) |
| | Gain-loss | 7.9 (1.4) | 6.0 (1.3) |
| | Sustained gain | 7.3 (1.6) | 5.1 (1.6) |

**Table 3** The incidence and ORs (95% CIs) of diabetes for different patterns of weight change over 10 years among urban residents in Japan

| Exposure variables | | Acquired DM/no. of subjects (incidence, %) | Crude | Multivariable |
|---|---|---|---|---|
| Men (no. for multivariate analysis=10 094) | | | | |
| Baseline age | Per 10 years | – | 1.44 (1.33 to 1.55) | 1.44 (1.29 to 1.61) |
| Weight change from 20 years of age, kg | < −5 | 7/1438 (0.5) | 2.18 (1.14 to 4.18) | 1.68 (0.75 to 3.80) |
| | −5 to +5 | 73/10 646 (0.7) | Ref | Ref |
| | +5 to +10 | 31/3200 (1.0) | 1.98 (1.45 to 2.72) | 1.54 (1.03 to 2.30) |
| | >+10 | 78/2166 (3.6) | 3.37 (2.54 to 4.48) | 2.08 (1.40 to 3.10) |
| Baseline BMI, kg/m² | <18.5 | 6/387 (1.6) | 1.35 (0.57 to 3.16) | 0.88 (0.29 to 2.70) |
| | 18.5–22 | 49/4236 (1.2) | Ref | Ref |
| | 22–25 | 168/7220 (2.3) | 2.04 (1.48 to 2.89) | 1.73 (1.14 to 2.63) |
| | >25 | 190/4699 (4.0) | 3.60 (2.62 to 4.94) | 2.52 (1.60 to 3.95) |
| Weight change pattern over 10 years | Sustained loss | 66/1903 (3.5) | 1.39 (1.03 to 1.87) | 1.11 (0.77 to 1.59) |
| | Loss-gain | 87/4644 (1.9) | 0.74 (0.56 to 0.97) | 0.63 (0.45 to 0.89) |
| | Stable | 142/5621 (2.5) | Ref | Ref |
| | Gain-loss | 38/3063 (1.2) | 0.49 (0.34 to 0.70) | 0.51 (0.32 to 0.82) |
| | Sustained gain | 80/1311 (6.1) | 2.51 (1.89 to 3.32) | 3.07 (2.15 to 4.39) |
| Smoking | None | 123/6385 (1.9) | Ref | Ref |
| | Ex-smoker | 150/5940 (2.5) | 1.32 (1.04 to 1.68) | 0.97 (0.71 to 1.31) |
| | Current-smoker | 140/4217 (3.3) | 1.75 (1.37 to 2.23) | 1.73 (1.25 to 2.34) |
| Alcohol drinking | None | 69/2250 (3.1) | Ref | Ref |
| | Sometimes | 48/1667 (2.9) | 0.94 (0.65 to 1.36) | 0.96 (0.66 to 1.40) |
| | Usually | 156/6177 (2.5) | 0.82 (0.61 to 1.09) | 0.78 (0.58 to 1.05) |
| Amount of walking per day | Per 30 min | — | 0.996 (0.92 to 1.08) | 1.01 (0.92 to 1.11) |
| Physical activity (sessions/week) | 0–1 | 85/3282 (2.6) | Ref | Ref |
| | 1–2 | 108/4105 (2.6) | 1.02 (0.76 to 1.36) | 1.02 (0.76 to 1.36) |
| | 3–5 | 43/1541 (2.8) | 1.08 (0.74 to 1.57) | 0.98 (0.67 to 1.45) |
| | 6–7 | 37/1166 (3.2) | 1.23 (0.83 to 1.82) | 1.06 (0.69 to 1.62) |
| Women (no. for multivariate analysis=10 614) | | | | |
| Baseline age | Per 10 years | — | 1.66 (1.48 to 1.87) | 1.78 (1.50 to 2.12) |
| Weight change from 20 years of age, kg | <−5 | 11/439 (2.5) | 0.71 (0.33 to 1.54) | 0.48 (0.18 to 1.25) |
| | −5 to +5 | 61/5243 (1.2) | Ref | Ref |
| | +5 to +10 | 109/4779 (2.3) | 1.42 (0.93 to 2.16) | 0.96 (0.55 to 1.65) |
| | >+10 | 232/6081 (3.8) | 5.41 (3.92 to 7.47) | 2.10 (1.20 to 3.67) |
| Baseline BMI, kg/m² | <18.5 | 20/2916 (0.7) | 1.24 (0.74 to 2.08) | 1.47 (0.76 to 2.86) |
| | 18.5–22 | 52/9388 (0.6) | Ref | Ref |
| | 22–25 | 60/3676 (1.6) | 2.98 (2.05 to 4.33) | 2.01 (1.21 to 3.34) |
| | >25 | 57/1470 (3.9) | 7.24 (4.95 to 10.59) | 2.91 (1.53 to 5.52) |
| Weight change pattern over 10 years | Sustained loss | 24/1661 (1.4) | 2.11 (1.25 to 3.54) | 1.48 (0.80 to 2.74) |
| | Loss-gain | 32/4022 (0.7) | 1.15 (0.72 to 1.86) | 0.72 (0.39 to 1.34) |
| | Stable | 36/5212 (1.3) | Ref | Ref |
| | Gain-loss | 30/4630 (0.6) | 0.94 (0.58 to 1.53) | 1.05 (0.57 to 1.95) |
| | Sustained gain | 67/1925 (3.5) | 5.19 (3.45 to 7.80) | 7.00 (4.11 to 11.94) |

Continued

**Table 3**   Continued

| Exposure variables | | Acquired DM/no. of subjects (incidence, %) | Crude | Multivariable |
|---|---|---|---|---|
| Smoking | None | 156/14,194 (1.1) | Ref | Ref |
| | Ex-smoker | 17/1789 (1.0) | 0.86 (0.52 to 1.1.43) | 0.85 (0.45 to 1.61) |
| | Current-smoker | 16/1467 (1.1) | 0.99 (0.59 to 1.67) | 1.20 (0.63 to 2.32) |
| Alcohol drinking | None | 78/5669 (1.4) | Ref | Ref |
| | Sometimes | 20/2035 (1.0) | 0.71 (0.43 to 1.17) | 0.91 (0.55 to 1.52) |
| | Usually | 27/2910 (0.9) | 0.67 (0.43 to 1.04) | 0.95 (0.60 to 1.51) |
| Amount of walking per day | Per 30 min | – | 1.01 (0.93 to 1.11) | 1.01 (0.90 to 1.13) |
| Physical activity (sessions/week) | 0–1 | 37/3894 (1.0) | Ref | Ref |
| | 1–2 | 42/3760 (1.1) | 1.18 (0.76 to 1.84) | 1.11 (0.70 to 1.75) |
| | 3–5 | 27/1874 (1.4) | 1.52 (0.93 to 2.52) | 1.14 (0.67 to 1.94) |
| | 6–7 | 19/1086 (1.7) | 1.86 (1.06 to 3.24) | 1.43 (0.79 to 2.60) |

BMI, body mass index; DM, diabetes mellitus; Ref, reference group.

intentional weight loss three or more times in 4 years carried ORs for the risk of diabetes of 1.11 (95% CI 0.89 to 1.37) and 1.39 (95% CI 0.90 to 2.13).[5] A study from a cohort of medical students at the Johns Hopkins University School of Medicine reported that the highest quartile of BMI variability for ages between 25 and 45 years had an OR of 2.1 (95% CI 1.0 to 4.6) for the risk of diabetes after 50 years of age compared with the other three quartiles.[21] In a large German cohort, weight cycling of ≥1.5 kg/year was significantly associated with an adjusted HR of 1.34.[6]

The disparity between the results of this study and the Western studies may be because of ethnic differences in diet,[22] the capacity to gain weight[23] and self-consciousness about bodyweight.[24] Further research is needed to explore why the relationship between weight cycling and risk of diabetes is reversed for Western and East Asian populations. It may be attributable to different motivations to lose weight because of different cultural views on dieting and body self-consciousness.[25] Those East Asians who try to lose weight may be more concerned about the poor health outcomes of being overweight. The Westerners in the studies who tried to lose weight may have been more likely to lose and regain a great deal of weight, and therefore potentially ran the risk of poor health outcomes.

The urban data in this study showed that an increase of >+5 kg above weight at 20 years of age increased the risk of developing diabetes with a dose–response relationship in both men and women (table 3). An increase of >+10 kg from weight at 20 years of age more than doubled the risk of diabetes compared with maintaining weight within ±5 kg. These results agree with the study involving a US cohort, which reported a relative risk of 3.2 (95% CI 1.4 to 7.4) for the highest quartile of an increase in BMI from 25 to 45 years of age in comparison with the other three quartiles.[21] A dose–response relationship of weight change from that at 20 years of age to the risk of diabetes was seen in urban Japanese women, with ORs of 0.48 (95%

CI 0.18 to 1.25) for a change of <5 kg, 0.96 (95% CI 0.55 to 1.65) for an increase of between 5 and 10 kg and 2.10 (95% CI 1.20 to 3.67) for an increase >10 kg (table 3). In contrast, Japanese men who lost ≥5 kg of bodyweight between the age of 20 years and early middle age had an OR of 1.68 (95% CI 0.75 to 3.80). This paradoxically increased OR was probably because of the small number of participants (seven) who developed diabetes among a weight loss group of 1438 people.

This study had several limitations. The first of these was the threshold of ±4% weight change in 10 years. This was drawn from a study in the UK on the association between weight change and the risk of diabetes.[26] It was also calculated as approximately one BMI unit in Japanese people with a mean BMI of 23 kg/m$^2$. However, the threshold for categorisation of weight cycling is likely to vary by mean BMI in different ethnicities because of different insulin sensitivities.[13 27] Second, we did not evaluate insulin sensitivity. Measuring fasting plasma glucose and insulin concentration to calculate HOMA-IR,[28] an index of insulin resistance, would have allowed us to assess the association between weight cycling and the physiological hazard of diabetes. Third, the weight changes recorded in the rural region may have been misclassified because of missing data from the years when participants did not undergo a medical check-up. However, this misclassification would have biased the ORs towards the null hypothesis, and we believe that this would therefore not change the conclusions for the rural region. Fourth, we could not examine whether weight cycling was intentional. We consider, however, that a subset of unintentional weight loss could be attributed to metabolic diseases, and patients with such diseases would not usually be able to regain the weight within a short time. Fifth, the follow-up duration and the numbers of weight measurements varied among the weight change groups. However, because perfect categorisation of weight changes over time would be impossible,

**Table 4** The incidence and ORs (95% CIs) of diabetes for different patterns of weight change over 10 years among rural residents in Japan

| Exposure variables | | Acquired DM/no. of subjects (incidence, %) | Crude | Multivariable |
|---|---|---|---|---|
| Men (no. for multivariate analysis=4818) | | | | |
| Baseline age | Per 10 years | – | 1.36 (1.08 to 1.72) | 1.60 (1.24 to 2.05) |
| Baseline BMI, kg/m² | <18.5 | 0/167 (0) | – | – |
| | 18.5–22 | 10/1425 (0.7) | Ref | Ref |
| | 22–25 | 21/2079 (1.0) | 1.44 (0.68 to 3.08) | 1.69 (0.79 to 3.63) |
| | >25 | 35/1148 (3.1) | 4.45 (2.19 to 9.03) | 5.81 (2.82 to 11.97) |
| Weight change pattern over 10 years | Sustained loss | 4/725 (0.6) | 0.43 (0.15 to 1.25) | 0.36 (0.12 to 1.05) |
| | Loss-gain | 13/681 (1.9) | 1.50 (0.75 to 3.00) | 1.58 (0.78 to 3.17) |
| | Stable | 22/1719 (1.3) | Ref | Ref |
| | Gain-loss | 4/916 (0.4) | 0.34 (0.12 to 0.99) | 0.44 (0.15 to 1.29) |
| | Sustained gain | 23/778 (3.0) | 2.35 (1.30 to 4.24) | 3.15 (1.70 to 5.83) |
| Smoking | None | 23/1695 (1.4) | Ref | Ref |
| | Ex-smoker | 8/998 (0.8) | 0.59 (0.26 to 1.32) | 0.72 (0.32 to 1.64) |
| | Current-smoker | 35/2126 (1.7) | 1.22 (0.72 to 2.07) | 1.50 (0.85 to 2.62) |
| Drinking | None | 11/1071 (1.0) | Ref | Ref |
| | Drinker | 55/3748 (1.5) | 1.44 (0.75 to 2.75) | 1.52 (0.79 to 2.94) |
| Women (no. for multivariate analysis=4852) | | | | |
| Baseline age | Per 10 years | – | 1.43 (0.99 to 2.06) | 1.23 (0.83 to 1.84) |
| Baseline BMI, kg/m² | <18.5 | 0/411 (0) | – | – |
| | 18.5–22 | 7/2135 (0.3) | Ref | Ref |
| | 22–25 | 14/1563 (0.9) | 2.75 (1.11 to 6.82) | 2.69 (1.07 to 6.75) |
| | >25 | 13/744 (1.8) | 5.41 (2.15 to 13.60) | 5.29 (2.07 to 13.51) |
| Weight change pattern over 10 years | Sustained loss | 3/863 (0.3) | 0.37 (0.11 to 1.29) | 0.32 (0.09 to 1.10) |
| | Loss-gain | 3/757 (0.4) | 0.42 (0.12 to 1.47) | 0.41 (0.12 to 1.44) |
| | Stable | 15/1615 (0.9) | Ref | Ref |
| | Gain-loss | 5/827 (0.6) | 0.65 (0.24 to 1.79) | 0.77 (0.28 to 2.14) |
| | Sustained gain | 8/791 (1.0) | 1.09 (0.46 to 2.58) | 1.43 (0.59 to 3.48) |
| Smoking | None | 2/646 (0.3) | Ref | Ref |
| | Ex-smoker | 29/3822 (0.8) | 2.45 (0.58 to 10.31) | 2.19 (0.52 to 9.26) |
| | Current-smoker | 3/375 (0.8) | 2.60 (0.43 to 15.60) | 3.33 (0.55 to 20.28) |
| Drinking | None | 31/3542 (0.9) | Ref | Ref |
| | Drinker | 3/1311 (0.2) | 0.26 (0.08 to 0.85) | 0.29 (0.09 to 0.95) |

BMI, body mass index; DM, diabetes mellitus; Ref, reference group.

we think that our study design enables us to answer the study question. Sixth, the urban data for weight at 20 years of age were self-reported by participants. They may therefore have been affected by recall bias. Seventh, a subset of the diagnoses in this study were not made by physicians but via an epidemiological criteria.[17] However, most observational studies rely on epidemiological criteria for detecting diabetes, and the use of a consistent diagnostic criterion allows researchers to compare the risk of diabetes onset between the reference group and groups of interest. Last, this study lacks any information about lifestyle, including diet, marital status, job type and car

ownership. All these may partly explain the association between weight cycling and diabetes.

This study also has several strengths. First, to the best of our knowledge, it is the first study to explore the relationship between weight cycling and the risk of diabetes in Asians. The relationship found in this study was almost directly opposite to that found in US studies, so further research in East Asians is necessary to confirm the findings. Next, this study was conducted in two different populations (urban and rural residents). The ORs of the risk of diabetes were 1.05 (95% CI 0.57 to 1.95) in the urban women with weight gain-loss (table 3) and 1.58

**Table 5** The incidence and ORs (95% CIs) of diabetes for different patterns of weight change over 10 years in middle-aged residents (45–64 years) in Japan

| Exposure variables | | Acquired DM/no. of subjects (incidence, %) | Crude | Multivariable |
|---|---|---|---|---|
| Urban middle-aged men (no. for multivariate analysis=4882) | | | | |
| Weight change pattern over 10 years | Sustained loss | 35/981 (3.6) | 1.04 (0.65 to 1.67) | 0.97 (0.60 to 1.55) |
| | Loss-gain | 62/2203 (2.8) | 0.79 (0.53 to 1.18) | 0.74 (0.49 to 1.11) |
| | Stable | 100/2749 (3.6) | Ref | Ref |
| | Gain-loss | 26/1098 (2.4) | 0.56 (0.32 to 0.999) | 0.57 (0.32 to 1.01) |
| | Sustained gain | 50/489 (10.2) | 3.09 (2.00 to 4.76) | 3.13 (2.00 to 4.89) |
| Urban middle-aged women (no. for multivariate analysis=5053) | | | | |
| Weight change pattern over 10 years | Sustained loss | 15/994 (1.5) | 1.30 (0.62 to 2.73) | 1.13 (0.53 to 2.41) |
| | Loss-gain | 20/1992 (1.0) | 0.87 (0.43 to 1.77) | 0.76 (0.37 to 1.57) |
| | Stable | 24/2285 (1.1) | Ref | Ref |
| | Gain-loss | 17/1642 (1.0) | 0.74 (0.34 to 1.62) | 0.80 (0.36 to 1.77) |
| | Sustained gain | 40/603 (6.6) | 5.62 (3.05 to 10.37) | 6.97 (3.67 to 13.25) |
| Rural middle-aged men (no. for multivariate analysis=2937) | | | | |
| Weight change pattern over 10 years | Sustained loss | 3/447 (0.7) | 0.48 (0.14 to 1.66) | 0.42 (0.12 to 1.47) |
| | Loss-gain | 11/449 (2.5) | 1.78 (0.81 to 3.91) | 1.76 (0.80 to 3.87) |
| | Stable | 15/1078 (1.4) | Ref | Ref |
| | Gain-loss | 4/546 (0.7) | 0.52 (0.17 to 1.58) | 0.58 (0.19 to 1.77) |
| | Sustained gain | 12/417 (2.9) | 2.10 (0.98 to 4.53) | 2.48 (1.12 to 5.49) |
| Rural middle-aged women (no. for multivariate analysis=3347) | | | | |
| Weight change pattern over 10 years | Sustained loss | 1/638 (0.2) | 0.16 (0.02 to 1.25) | 0.14 (0.02 to 1.01) |
| | Loss-gain | 3/579 (0.5) | 0.54 (0.15 to 1.92) | 0.54 (0.15 to 1.94) |
| | Stable | 11/1140 (1.0) | Ref | Ref |
| | Gain-loss | 4/558 (0.7) | 0.74 (0.24 to 2.34) | 0.82 (0.26 to 2.62) |
| | Sustained gain | 5/432 (1.2) | 1.20 (0.42 to 3.48) | 1.30 (0.44 to 3.84) |

BMI, body mass index; DM, diabetes mellitus; Ref, reference group.

(95% CI 0.78 to 3.17) in rural men with weight loss-gain (table 4). All other weight-cycling patterns for both sexes in both urban and rural regions were negatively correlated with the risk of diabetes, although some relationships were not statistically significant (tables 3 and 4). Third, the study included a large number of participants across both sexes (approximately 10 000 in the urban region and 5000 in the rural region). Its findings are therefore likely to be generalisable to Japanese and East Asian weight cyclers.

This study could help public health practitioners and on-site clinical professionals prevent diabetes in the general population. A sustained weight gain of >4% over 10 years in middle age (table 5) and >5 kg of additional weight gain since the age of 20 years (table 3) may both carry an increased risk of diabetes for both sexes. Middle-aged people can easily undergo such a small weight gain over the short or long term, so even non-diabetic people within the normal BMI range should be cautious about the risks resulting from small weight gains throughout their lifetime. An interventional study indicated that weight loss ($-1.8\,kg/m^2$ of BMI in

a diet intervention group and $-3.3\,kg/m^2$ of BMI in a diet-and-exercise intervention group) improved insulin sensitivity in Japanese patients with obesity and type 2 diabetes.[29] Improved insulin sensitivity was also observed in Americans who maintained their weight with treadmill-based exercise and no alteration in their diet.[30] Studies have also demonstrated that building muscle through exercising without changing weight improves insulin sensitivity.[31]

While not statistically significant, two weight-cycling patterns were found to result in an increased risk of diabetes (tables 3 and 4). Better insulin sensitivity leads directly to a decreased risk of diabetes, so these results may be because of differences in how the participants lost or gained weight (ie, whether they lost or gained fat or muscle mass). Recent studies have indicated that sarcopenia, the loss of skeletal muscle alone or with increased fat mass in ageing, is a leading cause of death in old age.[32] The results of our study, in the context of previous studies, suggest that over both the short and long term, people might reduce their risk of diabetes by losing fat and maintaining muscle mass.

## CONCLUSIONS

In urban men in Japan, weight cycling was associated with a significant reduction in the risk of diabetes. No clear association was seen in either urban women or in men and women living in the rural region. The results were different from those seen in Western countries and may be attributed to differences in diet, endocrinological capacity to gain weight and weight-consciousness. The risk of diabetes seems to increase linearly with weight gain from the age of 20 years in urban Japanese men and women. A study that includes the measurement of insulin sensitivity is necessary to confirm these results and to improve the understanding of the risks for East Asian weight cyclers.

**Acknowledgements** The authors thank all participants in this study and staff at the St. Luke's International Hospital and the Yamanashi Koseiren Health Care Center for their support, which made this study possible.

**Contributors** ZY, YY, MT and OT: setting up the study and data collection. HY, ZY, MM and AT: designing the study. HY and AT: data analysis. HY: writing and revising the draft. ZY, SO, MM, YA and HY: development of the discussion section. All authors read and approved the final manuscript.

**Funding** This work was supported by theMinistry of Education, Culture, Sports, Science and Technology of Japan (MEXT) (KAKENHI grant number: JP15K08730 and JP15K15221). The funder had no role in study design, analysis, decision topublish or preparation of the manuscript.

**Competing interests** None declared.

**Ethics approval** The ethics committee of the School of Medicine, University of Yamanashi (approval number: H27-1417)

**Provenance and peer review** Not commissioned; externally peer reviewed.

**Data sharing statement** No additional data are available.

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
