## [Reviewer comments · BMJ Open]

ARTICLE DETAILS

TITLE (PROVISIONAL)	Weight cycling and the subsequent onset of type 2 diabetes mellitus: 10-year cohort studies in urban and rural Japan
AUTHORS	Yokomichi, Hiroshi; Ohde, Sachiko; Takahashi, Osamu; Mochizuki, Mie; Takahashi, Atsunori; Yoda, Yoshioki; Tsuji, Masahiro; Akiyama, Yuka; Yamagata, Zentaro

VERSION 1 - REVIEW

REVIEWER	Tilman Kühn German Cancer Research Center (DKFZ) Heidelberg
REVIEW RETURNED	29-Oct-2016

GENERAL COMMENTS	Yokomichi and colleagues present data on weight development and diabetes risk in adults from two Japanese cohorts, the first located in a metropolitan area, the second one in a rural area. They show that weight cycling (defined by one cycle of either 4 % weight loss followed by 4 % weight gain or vice versa) is related to lower diabetes risk in specific subgroups. The study is very interesting, as previous similar studies mostly included North American participants with somewhat different cultural determinants of body weight. I have the following comments and questions: 1. The authors conclude that weight cycling may reduce the risk of diabetes. However, this result is not entirely consistent across the pre-defined subgroups in the study. Inverse risk associations with both weight cycling categories (up-down, down-up) are only significant in men from the urban region. So I am not sure if it can be concluded that weight cycling may be beneficial overall.2. It is not entirely clear how weight cycling was categorized. Could the authors add a scheme including number of weight assessments and time points? If weight cycles were defined based on weight data from two examinations, this should be discussed as a limitation, as only one cycle can be defined. How long was the duration between the last weight measurement and diagnosis?3. The age range of the populations is rather wide. Weight loss in the elderly may have a different meaning than in middle-aged people. Could the authors comment on heterogeneity by age?4. It would be helpful to have more descriptive data presented by weight development categories to get a feeling for background factors and confounders. Have the authors considered adding a supplementary table?
--

	5. Adjustment factors in the footnotes of all tables including odds ratios would help the reader. 6. In the discussion the authors state that their results are in contrast to those from studies in people with European descent that show higher risks with weight cycling. However, the two American studies [4,5] cited in the introduction have not clearly shown higher diabetes risk with weight cycling after adjustment. The evidence on weight cycling and diabetes risk from European and American studies is quite mixed, with some pointing to higher risks https://www.ncbi.nlm.nih.gov/pubmed/1342317 https://www.ncbi.nlm.nih.gov/pubmed/9080261 and others showing no clear associations https://www.ncbi.nlm.nih.gov/pubmed/14981219 https://www.ncbi.nlm.nih.gov/pubmed/20110286 https://www.ncbi.nlm.nih.gov/pubmed/26376796 7. The two cohorts in the study are presented separately. Could the authors explain, why data was not pooled? The population characteristics shown in table 1 look quite similar and both cohorts seem to consist of workers of certain companies, even if either from urban or rural areas. Given the rather low overall case numbers in some of the weight development categories pooling may make sense. Formal tests for interaction by study center (and also sex) in pooled analyses could be of interest, unless there is a good rationale for pre-defined subgroup analyses. 8. Could the cohorts be described in more detail in the methods? What types of jobs did the participants have, for example? Not all readers may go back to the studies' protocols. 9. Was there a possibility to exclude participants with unintentional weight loss from the analyses?
--	---

REVIEWER	Dr Lorna Aucott University of Aberdeen Aberdeen, Scotland Uk
REVIEW RETURNED	15-Nov-2016

GENERAL COMMENTS	This topic is needed but I'm concerned about the analysis and also about the interpretation of the result presented This paper explores the impact that weight changes particularly cycling might have on the risk of diabetes in Japan. The authors claim this has not previously been determined and also point out the debate about the association DM in Asian communities with BMI is thought to be at a lower BMI threshold compared to Western ones. There is an attempt to gain information about males and females and also a split depending whether they are rural or urban based. These are all reasonable points for this topic but I have concerns about the analyses and the interpretation.  • From the tables it would seem that the case:control ratio is
--

	small suggesting that OR's should estimate RR's reasonably. But the analyses does not account for time – it is questionable if those with DM will have developed it at the same rate. I would suggest if at all possible hazard ratios would be more appropriate as per the Framingham Study.  • However taking the results presented are of some concern. Other than Urban men none of the results are actually significant for weight cycling. None-less-the authors persist in interpreting all the OR's associated with cycling, stating that weight cycling causes a reduction in risk of DM... at best this could only be interpreted as 'men with diabetes were less likely to have experienced weight cycling'. • Weight change since baseline was reasonably (?) established over 3-10 years with at least 2 weight checks from annual medical check-ups (for the private companies employees – possibly for the Yamanashi Prefecture with private health care??). Weight change from the age of 20 years was also asked for but presumably as recall. Any risk of bias is not acknowledged in the limitations. • The title states 'weight change patterns' but the authors concentrate on weight cycling ignoring other results. This is a useful analyses but I would be nervous to advocate weight cycling as a 'protective' measure even if these analyses and results were more convincing. The discussion must consider this wider issue more fully.
--	---

VERSION 1 – AUTHOR RESPONSE

Responses to Reviewer #1

1. The authors conclude that weight cycling may reduce the risk of diabetes. However, this result is not entirely consistent across the pre-defined subgroups in the study. Inverse risk associations with both weight cycling categories (up-down, down-up) are only significant in men from the urban region. So I am not sure if it can be concluded that weight cycling may be beneficial overall.

We appreciate your advice to modify the conclusions. We have revised the conclusions as follows:

Line 38, Abstract:

Conclusions: In Japan, weight cycling is associated with a significant reduction in the risk of diabetes for men from an urban region and non-significantly for women from a rural region. However, the associations are unclear for women from the urban region and in men of the rural region. These results differ from those in Western studies, likely due to the motivations underlying weight cycling.

Line 309:

CONCLUSIONS

In men in the urban region and women of rural region in Japan, weight cycling has been associated with a reduction in the risk of diabetes, with statistical significance and non-significance, respectively; however, a clear association was not observed in the women of the urban region and men of the rural region. The results were different than those in Western countries and may be attributed to the motivations of the weight cyclers. In addition, the risk of diabetes increases linearly with weight gain from the age of 20 years in Japanese urban men and women. A study that includes the measurement of insulin sensitivity is necessary to confirm the present results and to improve the understanding of the risks for East Asian weight cyclers.

2. It is not entirely clear how weight cycling was categorized. Could the authors add a scheme including number of weight assessments and time points? If weight cycles were defined based on weight data from two examinations, this should be discussed as a limitation, as only one cycle can be defined. How long was the duration between the last weight measurement and diagnosis?

We appreciate your point that the explanation of categorisation of weight change patterns (exposure) and observation of the outcome (incidence or non-incidence of diabetes) were confusing. We have elaborated on the scheme of exposure to outcome. We hope the explanation can be understood clearly now. We have also added a scheme to explain exposure to outcome.

Line 85: Thus, the participants in the urban area received almost annual medical check-ups over 10 years, and those in the rural area received occasional voluntary check-ups. Participants were included in the analysis if they had no diagnosis of diabetes and an HbA1c less than 6.5% (48 mmol/mol) during a baseline period for the first three years of the 10-year period. If they received medical check-ups two or three times during the first three years, the data from the first visit were adopted for the baseline. Those included were also required to undergo a medical check-up at least twice during the latter seven years of the study. Hence, participants needed to receive at least three medical check-ups for the categorisation of weight change patterns (exposure).

Line 100:

Weight change categories

The participants were categorised into five groups according to their patterns of weight change during the 3–10 years since the baseline period. The stable group was comprised of the participants whose weight had not changed from the baseline by more than $\pm 4\%$. The sustained gain group consisted of the participants who gained more than 4% of their baseline weight and did not subsequently lose this extra weight. Similarly, the sustained loss group was comprised of participants who lost more than 4% of their baseline weight and did not subsequently regain this weight. The gain-loss group included participants who gained more than 4% of their baseline weight but brought their weight back below +4%. The loss-gain group included participants who lost more than 4% of their baseline weight but brought their weight back above -4%. From the last time point of this categorisation, the outcome of incident diabetes was measured. Therefore, the duration of observing whether the participants developed diabetes was between one and six years among the categories. The incidence of diabetes was measured after the participants were categorised, and any data measured after a diagnosis of diabetes were ignored to conserve the temporality of exposure to outcome for an epidemiological causation.¹⁹ The incidence of diabetes was measured after the participants were categorised, and any data measured after a diagnosis of diabetes were ignored to conserve the temporality of exposure to outcome for an epidemiological causation.

Figure 1. Scheme of how participants were categorised into the five weight change patterns

3. The age range of the populations is rather wide. Weight loss in the elderly may have a different meaning than in middle-aged people. Could the authors comment on heterogeneity by age?

Thank you very much for your insightful point. We agree with your point, and because one of our main focuses in this study was weight cycling in middle-aged adults during their most productive years, we have added a sensitivity analysis with the limitation of the middle-aged population.

Line 131: In addition, another focus of this study was the impact of weight cycling on incidental diabetes in middle-aged individuals. Hence, for a sensitivity analysis, we restricted the analyses to a

middle-aged population of 45 to 64 years.

Line 177:

Table 4 shows the risk of diabetes risk due to weight cycling in the middle-aged population. The ORs (95% CIs) for incidental diabetes were 0.57 (0.32–1.01) in the gain-loss group and 0.74 (0.49–1.11) in the loss-gain group among men living in the urban area. The corresponding ORs (95% CIs) were 0.80 (0.36–1.77) and 0.76 (0.37–1.57), respectively among women living in the urban area; 0.58 (0.19–1.77) and 1.76 (0.80–3.87), respectively among men living in rural area. The ORs were 0.82 (0.26–2.62) and 0.54 (0.15–1.94), respectively among the women living in the rural area.

Table 4. The incidence and odds ratios of diabetes related to patterns of weight change over 10 years in middle-aged residents (45–64 years) in Japan

Line 198: These results are reinforced by a sensitivity analysis with a restriction on middle-aged individuals, representing almost same ORs as those in the entire population except for the OR in the gain-loss group among women residing in the urban region.

4. It would be helpful to have more descriptive data presented by weight development categories to get a feeling for background factors and confounders. Have the authors considered adding a supplementary table?

We thank you for your suggestion to show all characteristics among the groups. We have added this information as a supplementary table.

Supplementary. Baseline characteristics among groups of weight change patterns over 10 years

5. Adjustment factors in the footnotes of all tables including odds ratios would help the reader.

Thank you for your suggestion to clarify the adjustment factors. We intended to show all the ORs in the multivariate analysis as 'adjusted odds ratios'. We have revised the manuscript to clarify that the adjusted ORs were those of the explanatory variables in the multivariate analysis.

Table 2. The incidence and odds ratios (95% CIs) of diabetes related to patterns of weight change over 10 years in residents of urban Japan

Table 3. The incidence and odds ratios (95% CIs) of diabetes related to patterns of weight change over 10 years in residents of rural Japan

6. In the discussion the authors state that their results are in contrast to those from studies in people with European descent that show higher risks with weight cycling. However, the two American studies [4,5] cited in the introduction have not clearly shown higher diabetes risk with weight cycling after adjustment. The evidence on weight cycling and diabetes risk from European and American studies is quite mixed, with some pointing to higher risks

<https://www.ncbi.nlm.nih.gov/pubmed/1342317>

<https://www.ncbi.nlm.nih.gov/pubmed/9080261>

and others showing no clear associations

<https://www.ncbi.nlm.nih.gov/pubmed/14981219>

<https://www.ncbi.nlm.nih.gov/pubmed/20110286>

<https://www.ncbi.nlm.nih.gov/pubmed/26376796>

We appreciate your advice to consider more literature concerning weight cycling and diabetes. We have reconsidered the Introduction and Discussion sections and added these studies.

Line 57: Researchers have also raised the question of whether repeatedly gaining and losing weight (weight cycling) is an independent risk factor for developing diabetes due to gaining weight. Studies on this topic are inconsistent in Westerners of Europe and the United States, and researchers currently appear to recognise that weight cycling may not independently increase but strongly induce the risk of diabetes. 4-8 In contrast, the risk of diabetes in Asian weight cyclers has not been researched.

Line 202: These observations are not consistent with those from previous studies of Western populations, which showed that weight cycling significantly or non-significantly increased the risk of diabetes for point estimates.

Line 214: In a large German cohort, weight cycling of ≥ 1.5 kg/year was significantly associated with an adjusted hazard ratio of 1.34.6

7. The two cohorts in the study are presented separately. Could the authors explain, why data was not pooled? The population characteristics shown in table 1 look quite similar and both cohorts seem to consist of workers of certain companies, even if either from urban or rural areas. Given the rather low overall case numbers in some of the weight development categories pooling may make sense. Formal tests for interaction by study center (and also sex) in pooled analyses could be of interest, unless there is a good rationale for pre-defined subgroup analyses.

Thank you very much for your suggestion. We have reconsidered the study design that you suggested. We analysed the urban and rural data separately because the lifestyles of the residents in the two regions differ. We hope that the additional sentences and literature will more clearly explain our analysis.

Line 68: In addition, the literature indicates that in Japan, diet, physical activity, prevalence of overweight individuals and aesthetic consciousness of metropolitan residents are different from those of rural residents.^{14 15} The aim of this study was to establish whether weight cyclers residing in Japan were at an increased risk of diabetes. We used Japanese urban and rural data to examine this clinical question in populations with varying lifestyles.

8. Could the cohorts be described in more detail in the methods? What types of jobs did the participants have, for example? Not all readers may go back to the studies' protocols.

Thank you for your helpful advice to describe the participants' jobs. We were not permitted to obtain occupation-related data. Instead, we have added a limitation concerning the lack of occupation data.

Line 268: Last, this study lacks the information pertaining to lifestyle, including diet, marriage status, job type and owning a car, that may have partly explain the association between weight cycling and incident diabetes.

9. Was there a possibility to exclude participants with unintentional weight loss from the analyses?

Thank you very much for your important point to consider weight cycling. We consider that for a

subset of the subjects, unintentional weight loss could have been due to metabolic disorders, and in most cases, patients with metabolic disease would not gain back weight over the short duration. However, we still recognise this is another limitation.

Line 259: Fourth, we could not examine whether the weight cycling was intentional; however, we consider that a subset of unintentional weight loss could be attributed to metabolic diseases, and patients with such diseases could not usually regain the weight within a short duration.

Answers to the comments of reviewer #2

10. From the tables it would seem that the case: control ratio is small suggesting that OR's should estimate RR's reasonably. But the analyses does not account for time – it is questionable if those with DM will have developed it at the same rate. I would suggest if at all possible hazard ratios would be more appropriate as per the Framingham Study.

Thank you very much for your insightful comment. Although we understand what you mean by 'rate' and 'accounting for time' in this study, the algorithm for determining the end of exposure and observed time of outcome (incident diabetes) is very complex because of missing values over ten-year-medical check-ups. In addition, the algorithm of this study design was very complicated. Thus, for this study, we would like to present odds ratios with logistic regression for adjusting confounding factors.

11. However taking the results presented are of some concern. Other than urban men none of the results are actually significant for weight cycling. None-less-the authors persist in interpreting all the OR's associated with cycling, stating that weight cycling causes a reduction in risk of DM... at best this could only be interpreted as 'men with diabetes were less likely to have experienced weight cycling'.

We appreciate your advice to carefully interpret the results. We have revised the conclusions as follows:

Line 38:

Conclusions: In Japan, weight cycling is associated with a significant reduction in the risk of diabetes for men from an urban region and non-significantly for women from a rural region. However, the associations are unclear for women from the urban region and in men of the rural region. These results differ from those in Western studies, likely due to the motivations underlying weight cycling.

Line 309:

CONCLUSIONS

In men in the urban region and women of rural region in Japan, weight cycling has been associated with a reduction in the risk of diabetes, with statistical significance and non-significance, respectively; however, a clear association was not observed in the women of the urban region and men of the rural region. The results were different than those in Western countries and may be attributed to the motivations of the weight cyclers. In addition, the risk of diabetes increases linearly with weight gain from the age of 20 years in Japanese urban men and women. A study that includes the measurement of insulin sensitivity is necessary to confirm the present results and to improve the understanding of the risks for East Asian weight cyclers.

12. Weight change since baseline was reasonably (?) established over 3-10 years with at least 2

weight checks from annual medical check-ups (for the private companies employees –possibly for the Yamanashi Prefecture with private health care??). Weight change from the age of 20 years was also asked for but presumably as recall. Any risk of bias is not acknowledged in the limitations.

We thank you for your advice to consider recall bias. We have added this as a limitation. Additionally, we would like to inform reviewer #2 that we, Japanese, can legally receive medical check-ups as residents and employees, and we sometimes pay for more comprehensive check-ups to detect cancer, arteriosclerosis, metabolic syndrome and osteoporosis.

Line 262: Fifth, the urban data for the weight at 20 years of age were derived from the participants' memory. Thus, recall bias could have existed in the results.

13. The title states 'weight change patterns' but the authors concentrate on weight cycling ignoring other results.

We have recognised from the references that the association between weight cycling and diabetes has been a topic in Western countries. We reconsidered that 'weight cycling' may be appropriate for this manuscript.

Title: Weight cycling and the subsequent onset of type 2 diabetes mellitus in urban and rural Japan

14. In this revision, we have modified some sentences for better readability.

We hope we have addressed the main points raised by the reviewers and the editor. Once again, we are extremely grateful for the insightful suggestions and hope that the manuscript is now acceptable for publication in BMJ Open.

VERSION 2 – REVIEW

REVIEWER	Tilman Kühn German Cancer Research Center, Heidelberg, Germany
REVIEW RETURNED	18-Jan-2017

GENERAL COMMENTS	Yokomichi and colleagues provide a revised and improved version of their very interesting manuscript on weight cycling and diabetes risk in Japanese cohorts from urban and rural regions. Their additional data and the new schema of weight change patterns are very helpful. I have one remaining question related to the categorization of weight changes: The number of examinations per participant was obviously different, and so were the time intervals between examinations. More importantly, the schema suggests that the number of examinations and time points was different across groups. E.g. sustained weight gainers only had 4 examinations (why is examination 2 not connected to the trend line by the way? What is depicted by the non-connected dots?) while people with sustained weight loss had six examinations. People with stable weight had four examinations in less follow-up time. I am not sure if such differential classification criteria are 100 % adequate, although I acknowledge that a perfect categorization or use of true weight trajectories based on multiple and continuous
---

	weight measurements in epidemiological studies is almost impossible. Could the authors comment on this in the discussion? The statement that "weight cycling may not independently increase but strongly induce the risk of diabetes" in the introduction is a bit confusing. I think it is sufficient to state that the evidence on weight cycling and risk of type 2 diabetes from prospective epidemiological studies is mixed, with a some studies poiting to a role of weight cycling while others don't.
--	--

VERSION 2 – AUTHOR RESPONSE

Answers to the comments of reviewer #1

4. Their additional data and the new schema of weight change patterns are very helpful. I have one remaining question related to the categorization of weight changes: The number of examinations per participant was obviously different, and so were the time intervals between examinations.

We appreciate your insightful suggestion about describing the variation in the number of weight measurements. We have added the duration of follow-up and the number of measurements for each group.

Line 155: For men in the urban region, means (SDs) of follow-up duration and numbers of measurements were 7.4 (1.9) years and 5.4 (1.6) in the sustained gain group, 8.2 (1.2) years and 5.7 (1.4) in the gain-loss group, 7.9 (1.7) years and 4.9 (1.7) in the stable group, 8.2 (1.2) years and 5.8 (1.4) in the loss-gain group and 7.7 (1.9) years and 5.8 (1.6) in the sustained loss group, respectively. The corresponding values for women in the urban region were 7.5 (1.9) years and 5.6 (1.6) in the sustained gain group, 8.2 (1.1) years and 5.9 (1.4) in the gain-loss group, 7.9 (1.6) years and 5.0 (1.7) in the stable group, 8.2 (1.2) years and 5.9 (1.4) in the loss-gain group and 7.9 (1.7) years and 6.0 (1.5) in the sustained loss group, respectively. For men in the rural region, means (SDs) of follow-up duration and numbers of measurement were 7.4 (1.6) years and 5.4 (1.6) in the sustained gain group, 7.8 (1.4) years and 6.1 (1.3) in the gain-loss group, 7.1 (1.9) years and 5.2 (1.6) in the stable group, 7.9 (1.4) years and 6.2 (1.2) in the loss-gain group and 7.3 (1.7) years and 5.2 (1.6) in the sustained loss group, respectively. The corresponding values for women in the rural region were 7.3 (1.6) years and 5.1 (1.6) in the sustained gain group, 7.9 (1.4) years and 6.0 (1.3) in the gain-loss group, 7.0 (1.8) years and 5.0 (1.6) in the stable group, 7.9 (1.3) years and 6.1 (1.3) in the loss-gain group and 7.4 (1.7) years and 5.4 (1.5) in the sustained loss group, respectively.

5. More importantly, the schema suggests that the number of examinations and time points was different across groups. E.g. sustained weight gainers only had 4 examinations (why is examination 2 not connected to the trend line by the way? What is depicted by the non-connected dots?) while people with sustained weight loss had six examinations. People with stable weight had four examinations in less follow-up time. I am not sure if such differential classification criteria are 100 % adequate, although I acknowledge that a perfect categorization or use of true weight trajectories based on multiple and continuous weight measurements in epidemiological studies is almost impossible. Could the authors comment on this in the discussion?

Thank you very much for your comments to clarify the details of the methodology and for pointing out another limitation of the study. In the baseline period of 3 years, the first health check-up was adopted. Therefore, the second and the third time points of receiving health check-up in the baseline duration were not connected by a line in the schema. Accordingly, the participants received health check-ups from a minimum of three to a maximum of eight times. We have clarified this point and

added another limitation to the study.

Line 88: If they received medical check-ups two or three times during the first three years, the data from the first visit were adopted for the baseline. Those included were also required to undergo a medical check-up at least twice during the latter seven years of the study. Hence, participants received three to eight medical check-ups to categorise weight change patterns (exposure).

Line 272: Fifth, the follow-up duration and the numbers of weight measurements vary among the groups of weight change patterns. However, because perfect categorisation of changing weight over time would be impossible, we think that this study design could answer the study question.

6. The statement that "weight cycling may not independently increase but strongly induce the risk of diabetes" in the introduction is a bit confusing. I think it is sufficient to state that the evidence on weight cycling and risk of type 2 diabetes from prospective epidemiological studies is mixed, with some studies pointing to a role of weight cycling while others don't.

Thank you for your suggestion to clarify a confusing sentence. We have revised it accordingly.

Line 58: Studies on this topic are inconsistent in Westerners of Europe and the United States; several prospective studies point to weight cycling as a risk factor for type 2 diabetes while others do not. 4-8

We hope we have addressed the main points raised by the editor and the reviewer. Once again, we are extremely grateful for the insightful suggestions and hope that the manuscript is now acceptable for publication in BMJ Open.